# Functional Analysis of Conserved Hypothetical Proteins from the Antarctic Bacterium, *Pedobacter cryoconitis* Strain BG5 Reveals Protein Cold Adaptation and Thermal Tolerance Strategies

**DOI:** 10.3390/microorganisms10081654

**Published:** 2022-08-16

**Authors:** Makdi Masnoddin, Clemente Michael Wong Vui Ling, Nur Athirah Yusof

**Affiliations:** 1Biotechnology Research Institute, Universiti Malaysia Sabah, Jalan UMS, Kota Kinabalu 88400, Sabah, Malaysia; 2Preparatory Centre for Science and Technology, Universiti Malaysia Sabah, Jalan UMS, Kota Kinabalu 88400, Sabah, Malaysia

**Keywords:** conserved hypothetical protein, quantitative PCR, Antarctic microbes, cold adaptation

## Abstract

*Pedobacter cryoconitis* BG5 is an obligate psychrophilic bacterium that was first isolated on King George Island, Antarctica. Over the last 50 years, the West Antarctic, including King George Island, has been one of the most rapidly warming places on Earth, hence making it an excellent area to measure the resilience of living species in warmed areas exposed to the constantly changing environment due to climate change. This bacterium encodes a genome of approximately 5694 protein-coding genes. However, 35% of the gene models for this species are found to be hypothetical proteins (HP). In this study, three conserved HP genes of *P. cryoconitis*, designated *pcbg5hp1*, *pcbg5hp2* and *pcbg5hp12*, were cloned and the proteins were expressed, purified and their functions and structures were evaluated. Real-time quantitative PCR analysis revealed that these genes were expressed constitutively, suggesting a potentially important role where the expression of these genes under an almost constant demand might have some regulatory functions in thermal stress tolerance. Functional analysis showed that these proteins maintained their activities at low and moderate temperatures. Meanwhile, a low citrate synthase aggregation at 43 °C in the presence of PCBG5HP1 suggested the characteristics of chaperone activity. Furthermore, our comparative structural analysis demonstrated that the HPs exhibited cold-adapted traits, most notably increased flexibility in their 3D structures compared to their counterparts. Concurrently, the presence of a disulphide bridge and aromatic clusters was attributed to PCBG5HP1’s unusual protein stability and chaperone activity. Thus, this suggested that the HPs examined in this study acquired strategies to maintain a balance between molecular stability and structural flexibility. Conclusively, this study has established the structure–function relationships of the HPs produced by *P. cryoconitis* and provided crucial experimental evidence indicating their importance in thermal stress response.

## 1. Introduction

In response to the extreme environment in the Antarctic, *Pedobacter cryoconitis* BG5, a psychrotolerant bacterium native to King George Island, Antarctica, produces protective proteins for survival and adaptation [1,2]. However, most of the proteins are designated as hypothetical, which shows no or very limited correlation to known proteins [3]. The limited information on hypothetical proteins (HPs) in extremophilic microorganisms proved to be an immense challenge in our effort to understand their cellular defence mechanisms [4]. To date, several investigations on many interesting discoveries about *P. cryoconitis* have been made regarding genes encoding resistance to cold stress and heavy metals, as well as industrial valuable enzymes [5]. However, despite the availability of the whole genome sequences, no detailed description of their stress response mechanisms has been documented [2]. This situation is further complicated by the fact that 35% of the protein-coding genes in *P. cryoconitis* genome are found to be hypothetical proteins [5]. These functionally unknown proteins may be involved in important aspects of this microorganism’s biological function. Previous research has demonstrated that a set of proteins with unknown functions is vital in the physiological regulation and cold adaptation of psychrophilic microorganisms [6,7]. Similarly, recent research suggests that the HPs in *P. cryoconitis* are important in the early stages of cold and freeze stress, though these findings have yet to be validated [8,9]. This presents an opportunity for discoveries to be made to gain a better understanding of their distinctive properties for adaptation mechanisms.

To understand how these proteins are involved in cold adaptation, it is important to determine the structural changes that are directly or indirectly involved in the adaptation to cold [10]. Functional annotation has been used to understand the molecular mechanisms of hypothetical proteins [11,12]. Studies that combine physicochemical properties with protein–protein interactions are crucial to predicting the structure, function and binding sites of these proteins [13]. Although psychrophiles share basic cold-adaptation strategies, different species have been shown to adopt different approaches to tolerating and surviving thermal stressors [14,15,16]. As every protein is unique, characterization at the physiological and biochemical level is critical for unravelling their molecular mechanisms and realising their full biotechnological potential. By elucidating the structure and function of the HPs, researchers will gain knowledge about new metabolic pathways and cascades, allowing us to better understand the protein mosaic and determine protein–protein interactions.

The present study was designed to evaluate the conserved HPs associated with thermal stress responses in *P. cryoconitis* BG5 to establish a better understanding of their possible adaptation mechanisms to thermal stress. While gene sequences provide important information, they are devoid of information on uncharacterized proteins, making it challenging to determine functionally significant sequences. Moreover, there is a limited number of studies addressing the properties of thermal stress proteins produced by psychrophilic microorganisms [1,8]. To date, there are no reports that focus on the functional and structural analysis of conserved HPs in *P. cryoconitis* BG5. To address this gap in knowledge about HPs, data from *P. cryoconitis* BG5 fully sequenced genomes are being utilized at the molecular level. Therefore, the objectives of this study are to elucidate the novel function and structure of HPs involved in thermal stress response in *P. cryoconitis* BG5, as well as to determine the relationship between protein molecular architecture and function. To accomplish these goals, methodologies from structural genomics and structural biology were integrated with omics technology. In vitro analysis of genes coding for HPs involved in thermal stress response was performed, including cloning, expression in *E. coli*, purification, and structural determination using comparative analysis. The findings from this study are intended to shed new light on the molecular structures adopted by biological molecules and provide insight into how different molecular architectures perform chemical reactions required for cell survival and adaptability.

## 2. Materials and Methods

### 2.1. Retrieval of Sequence Genome Data

The protein-coding genes were retrieved from the Rapid Annotation using Subsystem Technology (RAST) analysis data (unpublished data) of *Pedobacter cryoconitis* BG5. This strain was isolated on King George Island, Antarctica and its genome data revealed a total of 4922 predicted protein-coding genes, of which 3098 were assigned with predicted functions. The remaining 1738, or about 35% of the protein-coding genes, were classified as HPs since they have not been functionally characterized [5]. The selected conserved hypothetical protein sequences that were predicted to have thermal stress response (GenBank ID: MT670404, MT670405, and MT670415) were retrieved in FASTA format for further characterization.

### 2.2. Sequence Analysis

#### 2.2.1. Physicochemical Characterization

The physicochemical properties of the HPs in raw sequence format were determined using the ProtParam tool (http://web.expasy.org/protparam/, accessed on 21 April 2022) of ExPASy [17]. The parameters include the composition of amino acids, their molecular masses, and their theoretical isoelectric point (pI). The calculated isoelectric point (pI) will be useful for protein characterization because, at the protein’s pI, the surface of the protein is covered with charge, but the net charge of the protein is zero, hence making it stable and compact [18,19,20]. Hence, this gave information about the buffer pH for protein purification and storage buffer pH for the purified proteins.

#### 2.2.2. Subcellular Localization

To assign the location of the HPs in the cell, the proteins of interest were checked for transmembrane helices via TMHMM server version 2.0 [21,22] and signal peptides via SignalP-5.0 Server [23].

#### 2.2.3. Sequence Comparison

A similarity search was performed using the Basic Local Alignment Search Tool (BLAST) against the NCBI non-redundant UniProtKB/SwissProt sequences database [24] to find homologous proteins from related organisms that can be predicted to have the same function as the query protein [25].

#### 2.2.4. Function Prediction

InterProScan (https://www.ebi.ac.uk/interpro/about/interproscan/, accessed on 21 April 2022) was used to predict the presence of domains and important sites in any functional protein families [26,27]. Depending on what the constituent signatures represent, each InterPro entry was labelled with a “type”. A new entry type, homologous superfamily, has been added to the current list of types as part of InterPro’s release 65.0. Proteins in a homologous superfamily have a shared evolutionary history, as evidenced by structural similarities [26].

#### 2.2.5. Determination and Validation of Three-Dimensional Structures

The amino acid sequence of the HPs with soluble overexpression was converted from the original DNA sequence via ExPASy Translate Tool (https://web.expasy.org/translate/, accessed on 3 September 2021) and used as targets for homology modelling using the Phyre2 server (www.sbg.bio.ic.ac.uk/phyre2/, accessed on 3 September 2021) [28] and I-Tasser server (https://zhanggroup.org/I-TASSER/, accessed on 3 September 2021) [29]. The generated 3D models were then subjected to structure refinement using the ModRefiner webserver (https://zhanggroup.org/ModRefiner/, accessed on 10 October 2021). The energy-minimized structures were assessed with PROCHECK Ramachandran plots, VERIFY3D and ANOLEA-web. All the homology modelled proteins were superimposed with the template using UCSF Chimera 1.10 [30]. As the detection of structural similarities in proteins would give elucidation of the biochemical functions, the homology-modelled conserved hypothetical proteins were subjected to analysis of protein–protein interactions using the Protein Interactions Calculator (PIC) server (http://pic.mbu.iisc.ernet.in/, accessed on 16 January 2022) [31].

### 2.3. Cloning of the Genes Coding for the Selected HPs

To verify the accuracy of the predicted functions for the HPs from the *P. cryoconitis* BG5 genome, gene cloning was conducted to prepare a system for heterologous expression of the selected HPs in *E. coli* BL21 (DE3). The psychrophilic bacterium *P. cryoconitis* BG5 was obtained from the Biotechnology Research Institute, University Malaysia Sabah, Sabah, Malaysia. Briefly, a single colony of *P. cryoconitis* was inoculated in a starter culture of 5 mL Luria–Bertani (LB) broth (Merck, Kenilworth, NJ, USA) for 48 to 72 h at 20 °C with shaking at 200 rpm [2,32]. Subsequently, the culture was harvested, and total genomic DNA was isolated using the DNeasy DNA extraction kit (Qiagen, Germantown, MD, USA) as described by the manufacturer’s instructions. The extracted genomic DNA was then used for gene amplification and subsequent recombinant DNA construction. Genes coding for the selected HPs of *P. cryoconitis* BG5 HPs were identified and amplified using MyTaq^TM^ Red Mix DNA Polymerase (Bioline, London, UK) with a specific primer design for each of the selected genes (Appendix A). As a control for positive amplification, the primer pairs *gasgt1F* (5′-GACGACGACAAGATGTCCTCCAAGA-3′) and *gasgt1R* (5′-GAGGAGAAGCCCGGTCAAGCACCC-3′) targeting *gasgt1* gene [33] were used. The forward primer contains an additional overhang sequence of 5′-GGTGATGATGATGACAAG-insert-specific sequence-3′, whereas the reverse primer contains an overhang sequence of 5′-GGAGATGGGAAGTCATTA-insert-specific sequence-3′ to facilitate DNA sequencing procedures by creating specific 14–21 nucleotide single-stranded overhangs. The amplified genes were verified using PCR and DNA sequencing using LIC sequencing forward (5′-TAATACGACTCACTATAGGG-3′) and reverse (5′-GAGCGGATAACAATTTCACAGG-3′) primers. For cloning in the pLATE51 expression vector, 5′ and 3′ overhangs on the purified PCR templates were prepared according to the manufacturer’s instructions (Thermo Fisher, Waltham, MA, USA). Briefly, purified PCR products were mixed with a T4 DNA Polymerase, LIC Buffer and pLATE51 vector. The annealing mixture was incubated at room temperature for 5 min and directly used for bacterial cell transformation in the *E. coli* BL21(DE3) cells.

### 2.4. Protein Expression and Purification

Following the successful cloning of all HPs from *P. cryoconitis* BG5 into the *E. coli* BL21 (DE3) host, protein expression and purification processes of the HPs proteins were carried out to obtain high-quality pure protein samples. The starter culture of the positive transformant was grown in 500 mL LB broth and was supplemented with 100 μg/mL ampicillin. The culture was grown at 37 °C with shaking at 200 rpm to an OD_600_ of ~0.5–0.6. Subsequently, the cultures were induced by adding 0.1 mM isopropyl-β-D-thiogalactsidase (IPTG) (Sigma-Aldrich, St. Louis, MO, USA) and were incubated at 16 °C for 18 h with shaking at 200 rpm. Cells were then harvested by centrifugation at 4000× *g* for 20 min. Cells were resuspended in 20 mM Tris-HCl (pH 8), 150 mM NaCl and disrupted by sonication. After sonication, the cells were centrifuged at 10,000× *g* for 20 min. The recombinant protein was then purified from the supernatant (soluble fractions) via nickel-NTA affinity chromatography. The binding buffer comprised 50 mM Tris-HCl and 150 mM sodium chloride at pH 8.0, whereas the elution buffer comprised 50 mM Tris-HCl, 150 mM sodium chloride and 500 mM imidazole at pH 8.0. The purified proteins were further purified by gel filtration chromatography via Superdex^®^ HiLoad 10/300 GL gel filtration column (GE Healthcare, Chicago, IL, USA) with 20 mM Tris-HCl pH 8.0, 300 mM NaCl, and 2 mM EDTA as the equilibration buffer. The purified PCBG5HP1, PCBG5HP2, and PCBG5HP12 were analyzed for purity using 15% SDS-PAGE (Appendix A). The protein bands were visualized by Coomassie brilliant blue R250 staining (Nacalai, Kyoto, Japan).

### 2.5. Functional Determination

#### 2.5.1. Gene Expression Profile

Quantitative PCR analysis was carried out to determine the expression pattern of the genes coding for the selected HPs. *P. cryoconitis* culture was grown in a 10 mL Luria–Bertani (LB) broth (Thermo Fisher, Waltham, MA, USA) at 20 °C on a rotary incubator shaker at 200 rpm for 72 h until it reached the mid-log phase. Subsequently, the cultures were then exposed to different temperatures of −20 °C, 12 °C, and 20 °C. The cells were harvested at 10,000 rpm for 10 min after 6 h of exposure to each temperature and the pellet was used for RNA extraction. The total RNA of *P. cryoconitis* BG5 after exposure to different temperatures was extracted using GENEzol^TM^ Reagent (Geneaid, New Taipei City, Taiwan) according to the protocols outlined by [34]. The RT-qPCR reactions were carried out using a real-time thermal cycler, CFX96 (Bio-Rad, Hercules, CA, USA). The assays were run using SensiFAST SYBR^®^ No-ROX Kit (Bioline, London, UK). Each real-time PCR mixture contained 10 μL of 2× SensiFAST™ SYBR^®^ No-ROX One-Step mix (Bioline), 0.8 μL of 10 μM gene-specific forward and reverse primer, 0.2 μL reverse transcriptase, 0.4 μL RiboSafe RNase Inhibitor, and 1.0 µL of total RNA (about 1 µg). Specific primers were designed based on sequence data from the *P. cryoconitis* database. A total of three sets of primers, which included *pcbghp1* and *pcbghp2* and *pcbghp12*, were used in the reactions (Table 1).

The real-time cycler conditions were set as follows: reverse transcription step at 45 °C for 10 min for 1 cycle; an initial activation step at 95 °C for 2 min; 50 cycles of the denaturation step at 95 °C for 10 s; annealing at 60 °C for 10 s; and then exposure to 72 °C for 10 s. At the end of each run, the melting curves for the amplicons were determined by raising the temperature during fluorescence monitoring by 0.4 °C from 60 °C to 95 °C. A standard curve was constructed using 10-fold serial dilutions (100, 10, 1, 0.1, and 0.01 ng) of RNA amplified with *16S* reference gene and *pcbghp* primers. For each gene-specific primer, three independent replicates were performed, each technically repeated three times. The level of mRNA was measured in *P. cryoconitis* cells exposed to 30 °C heat treatment or cold shock at 4 °C for 6 h. The expression data were normalized against the *16S* transcript as an endogenous reference [5] to correct for differences in the starting amount of RNA and the efficiency of the reverse transcription reactions. The Ct value of gene expression collected through the amplification curves was used to analyze the relative expression level by using the comparative critical threshold Ct (2^−∆∆Ct^) method [35]. A one-way analysis of variance (ANOVA) at a significance level of *p* < 0.05 was used to compare the expression levels at different temperatures.

#### 2.5.2. Enzymatic ATP Hydrolysis

The ATPase activity was evaluated by the measurement of the colorimetric product resulting from the malachite green reagent and free ion [PO4]^3−^ measured at 620 nm. The ATPase assays were carried out using an ATPase/GTPase Activity Assay Kit (Sigma-Aldrich, St. Louis, MO, USA). The assay reaction mixture was composed of 10 µg recombinant protein (for the ATPase activity measurements) incubated in a 30 uL reaction volume containing 20 µL 40 mM Tris, 80 mM NaCl, 8 mM MgAc2, 1 mM EDTA at pH 7.5, and 10 uL 4 mM ATP (Sigma-Aldrich, St. Louis, MO, USA). To monitor the effect of temperature on the activity of PCBG5HP1, PCBG5HP2, and PCBG5HP12, the ATPase reaction mixture was incubated at room temperature or 4 °C for 30 min. After incubation, the reaction was stopped by adding 200 µL of malachite green reagent and incubated for an additional 30 min at room temperature to generate the colorimetric product. The product mixtures were loaded onto a 96-well plate and the absorbance values of colorimetric products were read using a SpectraMax spectrophotometer (Molecular Devices, San Jose, CA, USA) at 620 nm. All samples were run in triplicates. Phosphate standard values for colorimetric detection were prepared according to the manufacturer’s instructions.

#### 2.5.3. Inhibition of Citrate Synthase Thermal Aggregation

Citrate synthase, a model substrate, was used to test for the chaperone function of proteins in vitro. Citrate synthase (Sigma-Aldrich, St. Louis, MO, USA) was diluted with 40 mM HEPES (4-(2-hydroxyethyl)-1-piperazineethanesulfonic acid) KOH buffer at pH 7.5 to a final concentration of 150 nM in 1.0 mL cuvettes and heated at 43 °C with 300 nM of the purified recombinant proteins of PCBG5HP1, PCBG5HP2, and PCBG5HP12. Aggregation of citrate synthase was monitored by measuring turbidity at 320 nm with a SPECTRA Max PLUS spectrophotometer every 5 and 10 min for 1 h. Bovine serum albumin (BSA) was used at 150 nM to evaluate the non-specific protection of citrate synthase [36].

## 3. Results

### 3.1. Analysis of the Conserved Hypothetical Proteins from the P. cryoconitis BG5 Genome

The RAST analysis of the *P. cryoconitis* BG5 genome revealed three coding sequences annotated to conserved HPs associated with temperature stress response (Appendix A). *p**c**bg5hp1*, *pcbg5hp2*, and *pcbg5hp12* were amplified at the target sizes of 1023 bp, 882 bp, and 573 bp, respectively, followed by cloning and verification using sequencing (Appendix A). For total RNA samples, the extracted RNA was analyzed using gel electrophoresis to determine its integrity and the presence of *16S* and *23S* rRNA (Appendix A).

### 3.2. Prediction of Physicochemical Properties and Subcellular Localization

The physicochemical analysis in Table 2 shows that the PCBG5HP1, PCBG5HP2, and PCBG5HP12 proteins had molecular weight values of about 37 kDa, 32 kDa, and 23 kDa, respectively. The isoelectric point (pI) values calculated vary from 5.56 to 5.78, indicating that these HPs were acidic and negatively charged. This parameter is important in protein purification because it indicates the point at which the protein’s mobility is zero in an electro-focusing device, and hence the point at which the protein will be eluted [17,37]. The TMHMM 2.0 server indicated that none of the PCBG5HP1, PCBG5HP2, and PCBG5HP12 proteins featured predicted transmembrane helices, and the SignalP 5.0 server predicted that none of these proteins contained signal peptides. As a result, it was expected that these conserved HPs would be located in the cytoplasm.

### 3.3. Predicted Proteins with Adaptational Functions in Response to Temperature Stress

The shortlisted HPs were annotated for homologous proteins from related organisms using the BLASTp tool against the NCBI nonredundant UniProtKB/SwissProt sequences database. Table 3 showed low similarity to the protein in the database, where PCBG5HP1 and PCBG5HP2 resulted in 28% and 25% identity, respectively. PCBG5HP12, on the other hand, showed no significant similarity. As a result, this suggested that the selected HPs were still incompletely characterized [38,39].

For the prediction of a homologous protein family and functional domains via InterProScan, limited information was generated from known proteins in the database (Table 4). This demonstrated that the HPs in *P. cryoconitis* were still poorly understood and described, leaving room for discoveries. The three *P. cryoconitis* HPs were assigned to a distinct homologous superfamily, but only PCBG5HP1 consisted of domains and putative functions that could be attributed. PCBG5HP2 and PCBG5HP12 had domains but their functions were unknown or poorly defined, and were collectively referred to as DUFs (domain of the unknown function). PCBG5HP1 was predicted as an alcohol dehydrogenase protein family and NAD(P)-binding domain. Similarly, PCBG5HP1 was assigned to the NAD(P)-binding domain, yet the functions were unknown. PCBG5HP12 was predicted with DinB/YfiT-like putative metalloenzymes homologous superfamily, but no domain or any gene ontology (GO) terms were gathered. The data showed that PCBG5HP1 was projected to have a biological process for an oxidation–reduction process (GO: 0055114), molecular functions for zinc ion binding (GO: 0008270), and oxidoreductase activity (GO: 0016491), all of which were associated with diverse biological and molecular processes.

### 3.4. Three-Dimensional Structures Analysis

Figure 1, Figure 2 and Figure 3 show the predicted structure of PCBG5HP1, PCBG5HP2, and PCBG5HP12 proteins modelled by the Phyre2 server. Each model has a high confidence score despite a sequence identity of less than 30% similar to PDB structures, indicating that the folds were possibly correct, and accurate in the core (2–4 Å). PROCHECK analysis showed that each model had 100% amino acids in favoured and allowed regions. Furthermore, model verification using Verify3D showed that the constructed models obtained a positive score of more than 80%. Analysis using ANOLEA showed acceptable energy calculations at the atomic level in the protein model structure. In the PCBG5HP1, PCBG5HP2, and PCBG5HP12 models, each was shown to have multiple additional loops, therefore indicating the cold-adapted features of the proteins [40,41]. The superimposed PCBG5HP1 model and its template gave a significant RMSD value of 0.529 Å (Figure 1b). Similarly, the superimposed PCBG5HP2 and PCBG5HP12 models resulted in a significant RMSD value of 0.852 Å (Figure 2b) and 0.529 Å (Figure 3b), respectively. The intraprotein interactions analysis revealed that all models had lower hydrophobic interactions than their respective homologs. This suggested their structural flexibility, which enabled them to function at low temperatures [42].

Interestingly, the constructed model of PCBG5HP1 was predicted with a single disulphide linkage between Cys-95 and Cys-103 residues (Figure 4a), often an indicator of structural rigidity that is generally absent in psychrophilic proteins [42]. Despite this, the distance of the disulphide linkages in PCBG5HP1 was much longer, indicating a weaker interaction than that observed in the *Neurospora crassa* oxidoreductase (PDB 3M6I) homolog (Figure 4b).

### 3.5. Pcbg5hp1, Pcbg5hp2, and Pcbg5hp12 mRNA Expression in P. cryoconitis BG5 under Heat-Shock and Cold-Shock Conditions

The gene expressions of *pcbg5hp1*, *pcbg5hp2*, and *pcbg5hp12* mRNA in *P. cryoconitis* were measured when cells were exposed to temperatures below (4 °C) or above (30 °C) their optimal growth temperature (20 °C) for 6 h. The results indicated that heat treatment or cold shock had no significant effect on the level of mRNA for the three *P. cryoconitis* BG5 genes (Figure 5). The *pcbg5hp1* mRNA expression remained constant at all temperatures, but *pcbg5hp2* expression gave a slight increase by 1.1-fold at 30 °C. In addition, the *pcbg5hp12* expression increased slightly to 1.4-fold after a 6 h heat treatment at 30 °C and 1.1-fold after a 6 h cold shock at 4 °C. This suggested that these genes were expressed constitutively when cells were exposed to temperatures significantly below or above their optimum development temperature (20 °C).

### 3.6. ATPase Assay Analysis

ATPase activity was performed at 25 °C, the temperature of optimal activity for the determination of inorganic phosphate (Pi) using the malachite green method. To determine the cold tolerance ability of PCBG5HP1, PCBG5HP2, and PCBG5HP12 to hydrolyze ATP at a temperature lower than 25 °C, ATPase activity assays were conducted at 4 °C, representing cold stress. The rate of ATP hydrolysis was determined by the amount of free phosphate produced by the proteins following a 30 min incubation at 4 °C and 25 °C (room temperature) in the presence of ATP and malachite green reagent (Figure 6). At the lower temperatures (4 °C) and room temperature (25 °C), the ATPase activities of PCBG5HP1, PCBG5HP2, and PCBG5HP12 were not considerably different, indicating that the proteins remained active at room and low temperatures.

### 3.7. Thermal Unfolding Assay Analysis

Purified PCBG5HP1, PCBG5HP2, and PCBG5HP12 proteins were further evaluated for chaperone-like activity by measuring heat-induced aggregation of a non-native protein, citrate synthase, at 43 °C for 60 min (Figure 7). The results indicated that the addition of PCBG5HP1 was found to reduce citrate synthase thermal unfolding. This indicated that PCBG5HP1 could protect citrate synthase from heat-induced aggregation (for up to 40 min). The other proteins, PCBGHP2 and PCBGHP12, did not significantly protect citrate synthase against denaturation by heat (43 °C) for 60 min. On the other hand, PCBG5HP2 and PCBG5HP12 exhibited a significant degree of heat aggregation. This was compatible with the characteristics of cold-adapted proteins, which were unstable at elevated temperatures.

## 4. Discussion

This study is the first functional characterization of the conserved HPs from the Antarctic bacterium, *P. cryoconitis* strain BG5. Comprehensive data mining from the RAST analysis data revealed three conserved HPs (PCBG5HP1, PCBG5HP2, and PCBG5HP12) that were assigned to a temperature stress response. These proteins were selected primarily due to the limited information available, which therefore provided the opportunity for discoveries. Besides that, the selection was limited to sequences with 500–1500 bp in length because shorter sequences tend to make small peptides and longer sequences tend to form proteins that have many different domains, which complicated the characterization procedures [46,47]. The physicochemical characterization and subcellular localization analysis play an important role in the elucidation of proteins predicted with unknown function [48]. Identifying a protein’s localization in the cellular space contributes to the protein’s functional characterization, primarily because protein function is normally linked to its location [49,50,51]. The composition of amino acids can be used to predict subcellular localization due to evolutionary adaption to distinct subcellular sites [52,53]. In the current study, the PCBG5HP1, PCBG5HP2, and PCBG5HP12 proteins were predicted to be in the cytoplasm. Proteins in this cellular localization are involved in functional processes such as biosynthesis and transport, which contribute to the secretion of substrates or even other proteins [54]. Based on the GO analysis, PCBG5HP1 was annotated with oxidoreductase activity, which has been previously linked to a higher capacity for adaptation to low temperatures [55]. Increased oxygen solubility at low temperatures increases the risk of reactive oxygen species (ROS), and diverse strategies appear to be involved in preventing free radical damage [56]. Certain oxidoreductases are involved in the scavenging of free radicals and the removal of toxic chemicals that accumulate in living organisms under conditions of oxidative stress [57]. In relation to the effect of Antarctic climate change [58], oxidoreductase may also have a role in boosting energy metabolism in response to high-temperature stress, as revealed in a study of the Antarctic alga, *Chlorella* UMACC [59].

Our study on the *pcbg5hp1*, *pcbg5hp2*, and *pcbg5hp12* expression patterns showed that when *P. cryoconitis* BG5 cells were exposed to temperatures below or above their optimal growth temperature (20 °C), these genes were expressed constitutively. The constitutive gene expression profile suggested that the *pcbg5hp1*, *pcbg5hp2*, *and pcbg5hp12* genes were possibly involved in essential *P. cryoconitis* BG5 biological activities. Constitutive gene expression levels can provide higher fitness than induction expression in a rapidly changing environment such as in the Antarctic Peninsula [60]. In the wild-type *Arabidopsis*, constitutive expression of a ubiquitin-conjugating enzyme gene confers improved tolerance to water stress induced by sorbitol or soil drought [61], while constitutive expression of the UVH6 gene in the *Arabidopsis* mutant restores the plant heat tolerance [62]. Furthermore, the study of *Pseudomonas aeruginosa* shows a constitutive expression of DksA1 gene which is involved in the tolerance to H_2_O_2_-induced oxidative stress [63]. The fact that PCBG5HP1 was annotated as an oxidoreductase, PCBG5HP2 as a NAD(P)-binding domain, and PCBG5HP12 as a member of the metalloenzymes homologous superfamily suggested that they might be three of the housekeeping genes and play essential roles in *P. cryoconitis* mechanisms of adaptation to oxidative stress in low-temperature conditions [55]. This was supported by a previous transcriptomic analysis of an Antarctic yeast, *G. antarctica* PI12, which revealed that oxidative stress-related genes were expressed constitutively and abundantly rather than being induced by thermal stresses [8]. The constitutive expression of these genes was most likely a result of constant exposure of *P. cryoconitis* to cold temperatures in the Antarctic environment. These cold-adapted proteins may be involved in a metabolic pathway related to thermal stress. For instance, the methyl-transferase METTL21A enzyme has been implicated in the control of thermal stress proteins, HSP70s, and has been shown to impact HSP70s’ affinity for their client proteins [64]. It has previously been demonstrated that not all cold shock proteins (Csps) are cold-induced, and the expression patterns of other Csps remain unknown. For example, a study of the expression regulation of Csp genes in *E. coli* K-12 at low temperatures revealed that some Csps are expressed constitutively, and those different stress proteins were involved at different stages of stress [65]. Similarly, recent work has shown that several molecular chaperones, detoxifiers of ROS, and transcription and translation genes were expressed constitutively in *G. antarctica* PI12 to tolerate Antarctica’s fluctuating freezing temperatures [8].

At the protein level, the purified PCBG5HP1, PCBG5HP2, and PCBG5HP12 proteins were subjected to biochemical analysis to determine their activity and stability under various temperature conditions. At low temperature (4 °C) and room temperature (25 °C), the ATPase activities of PCBG5HP1, PCBG5HP2, and PCBG5HP12 were not significantly different. This indicated that protein activity was maintained at low and moderate temperatures. These findings shed light on the functional annotation of proteins containing a NAD(P)-binding domain. For both incubation temperatures, PCBG5HP12 produced the highest phosphate concentration. This might be related to the protein’s enzyme properties, since it has previously been annotated as a member of the DinB/YfiT-like putative metalloenzymes (IPR034660) homologous superfamily. These findings corroborated earlier research indicating that cold-adapted enzymes exhibited a shift in their optimum activity toward low temperatures and an increase in their specific activity at low and moderate temperatures [66]. This was because their high specific activity at active sites compensated for the exponential decrease in chemical reaction rates associated with low-temperature environments [67,68]. The thermal unfolding experiment indicated that PCBG5HP1 reduced the turbidity generated by citrate synthase when heated at 43 °C. This indicated the chaperone activity of the proteins and their ability to recognize and bind unfolded proteins in vitro, hence avoiding aggregation [69,70]. Similarly, it was shown that purified HSP20 from the hyper thermophilic archaeon *Sulfolobus solfataricus* P2 at concentrations of 150 nM or 300 nM efficiently protected citrate synthase against denaturation caused by heat treatment at 43 °C for 60 min [71]. On the other hand, PCBG5HP2 and PCBG5HP12 exhibited considerable heat aggregation. This implied that the proteins were cold-adapted, as evidenced by their low stability at high temperatures [72]. Cold-adapted enzymes’ thermal instability has previously been demonstrated using unambiguous methods such as fluorescence spectroscopy and differential scanning calorimetry [66,73]. For example, the thermal unfolding experiment of a cold-adapted DNA ligase from *Pseudoalteromonas*
*haloplanktis* revealed the protein’s low thermostability, as evidenced by the substantial transition temperature differences (Tmax) [66].

The determination of the 3D protein structures of PCBG5HP1, PCBG5HP2, and PCBG5HP12 was crucial to elucidate the relationship between their structures and functions, thus helping us to understand the mechanisms of thermal adaptation in cold-adapted proteins [41,74]. PCBG5HP1, PCBG5HP2, and PCBG5HP12 were all revealed to have numerous extra loops, therefore indicating the cold-adapted features of the proteins [40,41]. Intraprotein interactions analysis revealed that all the constructed models of *P. cryoconitis* BG5 have a lower hydrophobic interaction than their respective homologs. This demonstrates their structural flexibility, which enabled them to function at low temperatures [42]. The enhanced flexibility most likely permitted the ATP binding site to undergo conformational changes during protein folding by reducing the activation energy of intra and inter-ring interactions between the subunits [75]. The findings were corroborated by a previous study on cold-adapted proteins from *Arthrobacter* sp. 32cB that demonstrated that the environment’s lack of free energy, caused by low temperature, can be compensated for by a higher efficiency of energy gain due to their high structural flexibility [74]. Interestingly, PCBG5HP1 possessed a single disulphide linkage between Cys-95 and Cys-103 residues. A disulphide bridge is often an indicator of structural rigidity that is generally absent in psychrophilic proteins [42]. This has been demonstrated by the analysis of the catalytic domain of chitobiase from a psychrophilic Antarctic bacterium, *Arthrobacter* sp. TAD20. The catalytic domain’s lack of disulphide linkage contributes to their conformational flexibility for cold-adapted activity [76]. However, oxidoreductases are known to have a role in bacterial pathogenicity by forming disulphide linkages, which maintain the stability and rigidity of numerous extracellular proteins [77]. This was possibly correlated with the ability of PCBG5HP1 to protect the citrate synthase against heat aggregation in the thermal unfolding assay. In contrast, the thermal stability of both PCBG5HP2 and PCBG5HP12 was decreased due to the low intraprotein interactions, particularly the aromatic–aromatic interactions. As psychrophilic enzymes have lower thermal stability than mesophilic enzymes, their apparent optima are shifted by 10 to 20 degrees Celsius toward low temperatures [68]. This was demonstrated in the thermal unfolding experiment, where PCBG5HP2 exhibited higher protein aggregation than the citrate synthase control reagent. The structural flexibility of psychrophilic proteins enabled the proteins to move more freely at low temperatures [72]. As a result of the increased mobility of the protein structure, it exhibited poor stability at high temperatures. The thermal unfolding assay of the cold-adapted DNA ligases from *Pseudoalteromonas haloplanktis* demonstrated that their thermolability was entropically driven, in which at any specific temperature, more intra-protein interactions are broken as compared to their mesophilic counterparts [78]. Despite this, in comparison to the *Neurospora crassa* oxidoreductase (PDB 3M6I) homolog, the distance between Cys-95 and Cys-103 disulphide linkage in PCBG5HP1 was much longer and weaker. Consequently, as demonstrated by the ATPase assay employing malachite green substrates, PCBG5HP1 was capable of maintaining its ATP hydrolysis reaction at both low and moderate temperatures. This was similar to a previous study on cold-active chitinase from *Arthrobacter* sp. TAD20, which indicated that the addition of a disulphide bridge enhanced stability while maintaining the enzyme’s psychrophilic characteristics [79]. This is consistent with the increased catalytic efficiency at lower temperatures that has been measured in certain psychrophilic enzymes when compared to their mesophilic or thermophilic homologs [80,81]. The site-directed mutagenesis experiment on psychrophilic α-amylase demonstrated that the disulphide bridge mutant exhibits thermodynamic stability comparable to that of the mesophilic enzyme while increasing the enzyme’s affinity for its substrate by a factor of almost four [82]. This is supported by the thioredoxin and thioredoxin reductase from the psychrophilic eubacterium *P. haloplanktis* that display a full activity at low temperatures but also exhibit exceptional heat resistance due to the presence of a functional disulphide bridge [83]. Previously, a comparative analysis of the TRiC chaperonin from *G. antarctica* suggested that the residue substitutions observed in the psychrophilic structure were subjected to positive selection or strategic design selection to avoid the loss of native configuration due to cold denaturation [84]. This signifies that the protein’s structural design was optimized for molecular stability while maintaining structural flexibility. This is supported by diffraction experiments conducted at −173.15 °C and room temperature, which revealed the structure of a cold-adapted β-D-galactosidase from the Antarctic bacterium *Arthrobacter* sp. 32cB, which maintains a balance between molecular stability and structural flexibility, which can be observed depending on the temperature at which the X-ray diffraction experiments are conducted [74].

## 5. Conclusions

The functional analysis of the three *P. cryoconitis* HPs (PCBG5HP1, PCBG5HP2, and PCBG5HP12) indicated cold-adapted traits, most notably increased flexibility in comparison to their mesophilic or thermophilic counterparts. This is mostly owing to their lower hydrophobic interactions and extra loop attributes, which have been related to increased flexibility for cold-adapted activities. The expression pattern at temperatures below and above the optimal growth temperature indicated constitutive expression of the genes possibly reflecting the essential roles of these genes in *P. cryoconitis* biological activities. It is suggested that *pcbghp1*, *pcbghp2* and *pcbghp12* might be three of the housekeeping genes in *P. cryoconitis*. Since housekeeping genes are required in basic cellular functions and maintenance, they are thus expected to maintain consistent expression levels across cells and conditions. The structural dynamics of the HPs were corroborated by the functional experiments, which revealed that all HPs exhibited ATPase activity at low and moderate temperatures, demonstrating their cold-adapted property. However, PCBG5HP1 demonstrated increased stability and protection against the thermal unfolding of citrate synthase at a high temperature of 43 °C. The existence of a disulphide linkage in PCBG5HP1 has been linked to this remarkable stability. Thus, it is perfectly feasible to elucidate that the HPs examined in this work adopt strategies to maintain a balance between molecular stability and structural flexibility.

## Figures and Tables

**Figure 1 microorganisms-10-01654-f001:**
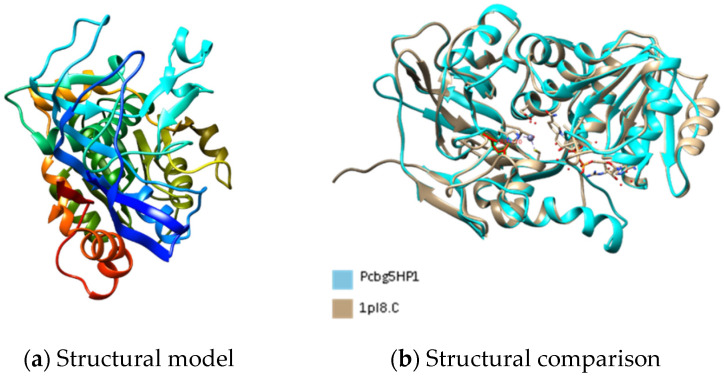
The three-dimensional structure analysis of PCBG5HP1. (**a**) The structural model of PCBG5HP1 was constructed using the Phyre2 server with 24% sequence identity to the structures of 1PL8 as templates [43]. Image coloured by rainbow N → C terminus. (**b**) Comparative structural analysis between PCBG5HP1 model and human sorbitol dehydrogenase/NADH/inhibitor complex (PDB 1PL8).

**Figure 2 microorganisms-10-01654-f002:**
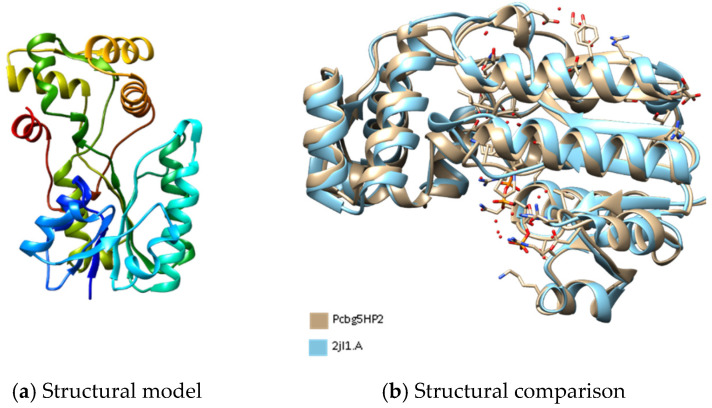
The three-dimensional structure analysis of PCBG5HP2. (**a**) The structural model of PCBG5HP2 was constructed using the Phyre2 server with 17% sequence identity to the structures of 2JL1 as templates [44]. Image coloured by rainbow N → C terminus. (**b**) Comparative structural analysis between PCBG5HP1 model and *Citrobacter* sp. triphenylmethane reductase (PDB 2JL1).

**Figure 3 microorganisms-10-01654-f003:**
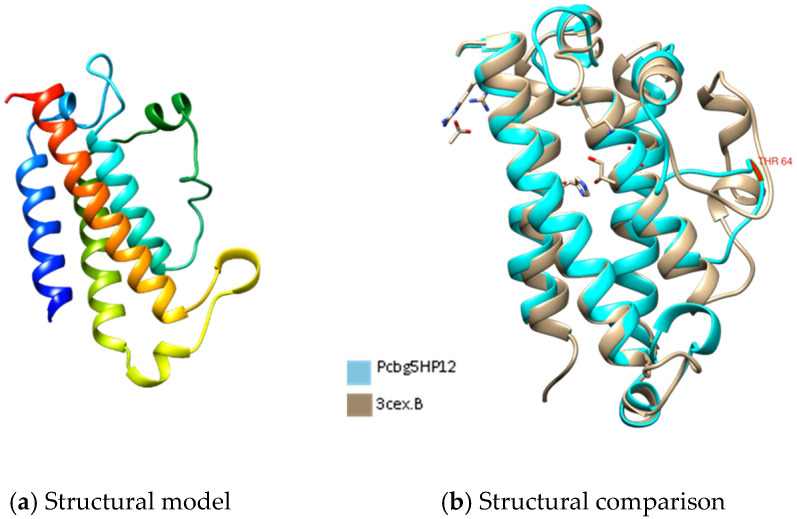
The three-dimensional structure analysis of PCBG5HP12. (**a**) The structural model of PCBG5HP12 was constructed using the Phyre2 server with 14% sequence identity to the structures of 3CEX as templates [45]. Image coloured by rainbow N → C terminus. (**b**) Comparative structural analysis between PCBG5HP1 model and *Enterococcus faecalis* uncharacterized protein (PDB 1PL8).

**Figure 4 microorganisms-10-01654-f004:**
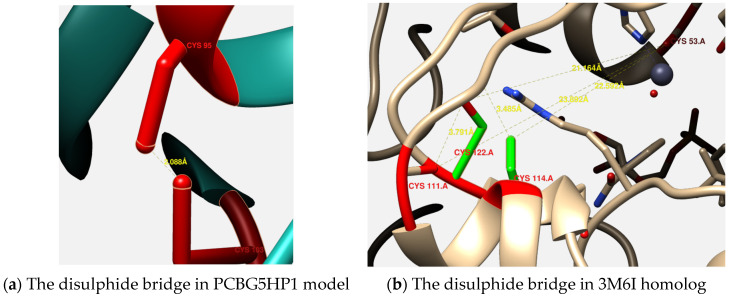
Comparison of the disulphide bridge in PCBG5HP1 and its homolog 3M6I. (**a**) The disulphide bridge between Cys-95 and Cys-103 in PCBG5HP1. (**b**) The disulphide bridge between Cys-53 and Cys-122, Cys-111, and Cys-114 in 3M6I homolog.

**Figure 5 microorganisms-10-01654-f005:**
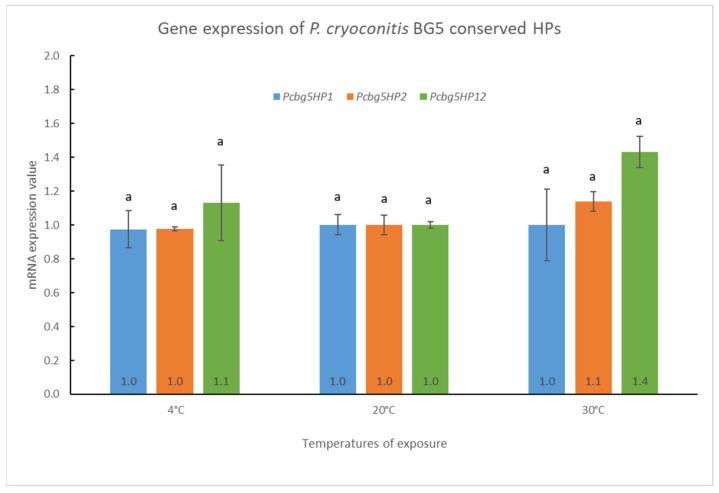
Gene expression analysis of *pcbg5hp1*, *pcbg5hp2,* and *pcbg5hp12.* Data were representative of three biological replicates and were expressed as the mean ± SD. The letter above columns indicated levels of different significance, *p* > 0.05. The level of mRNA expression was measured in cells cultured at the indicated growth temperature for 6 h and normalized against *16S* rRNA gene (internal control) levels. The mRNA expression value at 20 °C was set to 1 and the other values were normalized against this.

**Figure 6 microorganisms-10-01654-f006:**
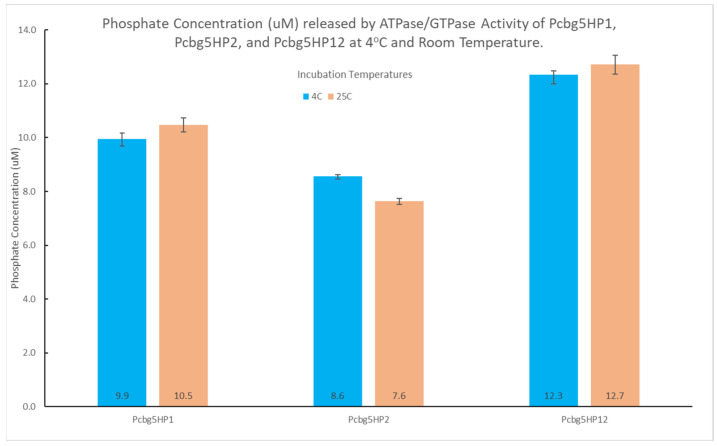
ATPase hydrolysis activity of PCBG5HP1, PCBG5HP2, and PCBG5HP12 recombinant proteins at 4 °C and 25 °C. Data were representative of three biological replicates and were expressed as the mean ± SD.

**Figure 7 microorganisms-10-01654-f007:**
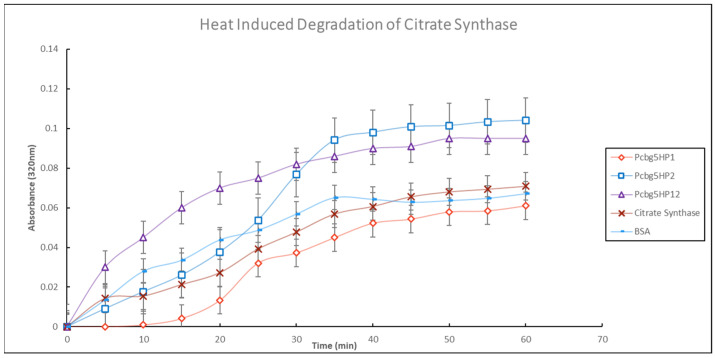
Heat-induced denaturation of citrate synthase. A total of 300 nM recombinant proteins of *P. cryoconitis* BG5 were purified and incubated at 43 °C with 150 nM citrate synthase, and turbidity was measured at 320 nm. Data were representative of three biological replicates and were expressed as the mean ± SD. BSA at 300 nM was a control sample for citrate synthase non-specific protection.

**Table 1 microorganisms-10-01654-t001:** Primer sequences for qPCR analysis of *P. cryoconitis* genes.

Gene Target	Direction	Primer DNA Sequence	Amplicon Length (Bp)
*pcbghp1*	Forward	5′-GGATATCCAGCTCAAAGAAAG-3′	96
	Reverse	5′-AGGTCTGTTCCGCAAATA-3′	
*pcbghp2*	Forward	5′-CAGAGTAGCGGTACAAATAAG-3,	119
	Reverse	5′-GACTAAGATCAACAACTCTGG-3,	
*pcbghp12*	Forward	5′-AAAGGATTGGCGTAACCGGG-3′	165
	Reverse	5′-GAGTGCCCCTGATTCCTGATG-3′	

**Table 2 microorganisms-10-01654-t002:** The physicochemical properties of *P. cryoconitis* HPs retrieved by the Protparam tool.

ID	Physicochemical Properties		
	Number of Amino Acids	Molecular Weight (Da)	Isoelectric Point (pI)
PCBG5HP1	340	37,301.82	5.56
PCBG5HP2	293	31,926.54	5.78
PCBG5HP12	190	22,542.34	5.77

**Table 3 microorganisms-10-01654-t003:** The BLAST search results for the selected HPs from *P. cryoconitis* against the NCBI non-redundant UniProtKB/SwissProt sequences database.

Protein ID	Sequence Identity	e Value	Description
PCBG5HP1	28%	5 × 10^−29^	O35045.1 Uncharacterized zinc-type alcohol dehydrogenase-like protein YjmD (*Bacillus subtilis* subsp. *subtilis* str. 168)
PCBG5HP2	25%	2 × 10^−8^	Q54LW0.2 Prestalk A differentiation protein A (*Dictyostelium discoideum*)
PCBG5HP12	-	-	No significant similarity found

**Table 4 microorganisms-10-01654-t004:** Functional annotation of *P. cryoconitis* HPs by the InterProScan tool.

Protein ID	Homologous Superfamily	Domains	Biological Process	Molecular Function	Cellular Component
PCBG5HP1	GroES-like superfamily (IPR011032) NAD(P)-binding domain superfamily (IPR036291)	Alcohol dehydrogenase, N-terminal (IPR013154) Glucose dehydrogenase, C-terminal (IPR031640)	GO:0055114 oxidation-reduction process	GO:0008270 zinc ion binding GO:0016491oxidoreductase activity	None predicted
PCBG5HP2	NAD(P)-binding domain superfamily (IPR036291)	NAD(P)-binding domain (IPR016040)	None predicted	None predicted	None predicted
PCBG5HP12	DinB/YfiT-like putative metalloenzymes (IPR034660)	None predicted	None predicted	None predicted	None predicted

## Data Availability

Not applicable.

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
