# Peer review of "Functional Analysis of Conserved Hypothetical Proteins from the Antarctic Bacterium, Pedobacter cryoconitis Strain BG5 Reveals Protein Cold Adaptation and Thermal Tolerance Strategies"

_microorganisms, 2022, doi:10.3390/microorganisms10081654_

Round 1
Reviewer 1 Report
The ms by Masnoddin et al describes the analysis of three conserved hypothetical proteins from an Antarctic bacterium.
In my opinion the manuscript cannot be accepted for publication in Microorganisms in its present form. However, the ms itself can be reconsidered for publication once the authors have responded adequately to all the issues raised by this referee.
Abstract
line 18: genes purified? Henes have not been purified; eventually the gene products can be purified.
line 20: the finding that the three genes are consititutively expressed doe not per se imply that they are involved in cellular processes associated with thermal stress. The authors should clarify and/or modify the statement.
line 27: demonstrated: replace with suggested
Introduction
line 46: significant aspects is not correct; replace with important aspects
line 63: replace protein pathways with metabolic pathways
line 67: of their possible adaptation
lines 79-81: HPs are not cloned and expressed; the genes coding for HPs are cloned and expressed
MATERIALS AND METHODS
2.3 Did the authors checked the nucleotide sequence of the amplified genes once thay have cloned in the (plasmid ?) vector. Even though in lines 154-156 they stated that the genes "were verified" they should rephrase the statement.
2.4 The authors did not describe the cloning vector they used; I imagine that they used a plasmid vector. If this is so, they should describe the cloning procedure and indicate whether the plasmid is an expression vector.
In addition to this they they state that the .. selected recombinant proteins... If the proteins are recombinant (as they stated) they should have been fused to other proteins. Please, clarify/modify this sentence.
2.5 It is not clear to me what culture medium was used for growing the Antarctic bacterium and at which growth phase the RNA was extracted.
RESULTS
line 243 of about 37 Kda
line 245: replace significant with important
line 255: the authors state that Table 3 shows a low similarity, but in table 3 the sequence identity was reported. Please, clarify. In addition to this it should be stated a low degree of sequence similarity (or identity)
lines 258-259, in my opini, this statement is not correct; indeed many orthologus proteins from different bacterial genera/species share a degree of sequence identity much lower than 90%. In addition to this, homologues is not a correct term, since homology can be referred to either paralogous or orthologus genes (or proteins). Hence if the authors refer to proteins sharing the same function. they should refer to orthologoues, since an extremely low number of paralogous genes with the same function si known in the bacterial domani (and this is restricted to gene coding for 16S rRNA).
Line 277: The fact that a given gene might be involved in various biological and molecular processes (that , in turn, should be demonstrated by in vivo experiments) doe not per se imply that it play a vital role, i.e. is essential for the bacterial life. The identification of essential genes in a given genome requires a Tnseq experiment or other experimental approaches.
Line 295: replace demonstrated with suggested
Lines 321-323: The authors do not describe the medium in which the bacterium was grown nor at which growth stage they extracted the RNA. They should also test different culture media; indeed different growth conditions might affect the level of mRNA synthesis. In addition to this, it is clear that if the authors used a maximum broth, this very likely imply that the environmental conditions in which the bacterium gros are completely different from the in vitro analysis. Hence, they should use a medium mimicking the original environment. For Antarctic bacteria, it has been set up a minimal medium (see works by Ermenegilda Parrili et al.).
Lines 340 and 343; it is not clear to me why the authors used room temperature (25°C) and not 20°C (i.e. the temperature at which they extracted the RNA).
Line 358: why recombinant? see also line 421
Lines 374-376; the authors state that ..longer sequences tend to have many different domains. Nucleotide sequences do not have domains, eventually they code protein domanis. Please rephrase the statement.
Lines 398 and 401: essential biological activities... sential roles (see above)
Lines 512-515; Again, the finding that the three genes are costitutively expressed (under which conditions?) does not per se inply that they play an essentia role in biological activities.
References
The authors have not checked adequately the references before submitting the manuscript. In some cases, the reference lacks the journal name (see for instance reference 13). In many other cases they have used different abbreviations for the same journal (references 24 and 25, references 39 and 55, references 53 and 68)
Author Response
We would like to thank the reviewers for their careful and thorough reading of this manuscript and for the thoughtful comments and constructive suggestions, which help to improve the quality of this manuscript. We agree with all their comments and we have revised our manuscript accordingly. Moreover, we have included all reviewers’ suggestions and clarified the text when needed. We are confident that the new version of the manuscript is greatly improved. We respond below in detail to each of the reviewer’s comments. We hope that the reviewers will find our responses to their comments satisfactory, and we are open to any further suggestions that the reviewers may have. Attached, please find the referees’ comments in italics and our responses inserted after each comment. Looking forward to hearing from you soon.

Reviewer 2 Report
The paper concerns investigation of hypothetical proteins (HPs) in extremophilic microorganism and its role in their adaptation mechanisms to thermal stress.
The paper is designed and written well. The introduction provides sufficient information about the investigated problem. Authors have used modern and well-chosen methods that are adequately described. The results are clearly presented and interesting.
Author Response
We would like to thank the reviewers for their careful and thorough reading of this manuscript and for the thoughtful comments and constructive suggestions, which help to improve the quality of this manuscript. We agree with all their comments and we have revised our manuscript accordingly. Moreover, we have included all reviewers’ suggestions and clarified the text when needed. We are confident that the new version of the manuscript is greatly improved. We respond below in detail to each of the reviewer’s comments. We hope that the reviewers will find our responses to their comments satisfactory, and we are open to any further suggestions that the reviewers may have. Attached, please find the referees’ comments and our responses inserted after each comment. Looking forward to hearing from you soon.

Round 2
Reviewer 1 Report
The revised version of the manuscript is improved in respect to the previous one.
However, some changes have to be made before acceptance for publication in Microorganisms. In detail:
line 18: delete protein bewteen their an functions
line 21: in some regulatory functions
line 190: -20°C, 12°C, and 20°C
Table 2: No oif amino acids
lines 275-276: Delete this sentence, since it is a repetition of the sentence at lines 270-271
lines 554-555: The constant expression profile of these genes possible indicates these genes possibly are the housekeeping genes that are required to maintain cellular function in P. cryoconitis
This sentence should be modified: indeed, the statement ...are the housekeeping genes that are required to maintain cellular function in P. cryoconitis... implies, in this form, that these genes are the only ones involved in maintaining cellular function.... This has not been demonstrated by the authors. They can only suggest that these genes might be three of the housekeeping genes.
In addition to this, they state that the constant expression profile. Constant refers to time; but the authors did not perform a dynamics (in terms of time) of the expression profiles. Hence, the sentence should be rephrased.
line 684; maybe Methods? and not methods
line 699: PLoS One or PloS ONE? (see also lines 636, 725 and 748)
Author Response
We would like to thank Reviewer-1 for his/her thoughtful comments and suggestions. We truly appreciate the time and efforts spent by Reviewer-1. We have amended our manuscript as suggested and corrected our grammatical errors including our list of references. We are confident that the new version of the manuscript is greatly improvised. Please find the detail of each of the comments in the attached document.
